# Green and Efficient Extraction of Polysaccharide and Ginsenoside from American Ginseng (*Panax quinquefolius* L.) by Deep Eutectic Solvent Extraction and Aqueous Two-Phase System

**DOI:** 10.3390/molecules27103132

**Published:** 2022-05-13

**Authors:** Rong-Rong Zhou, Jian-Hua Huang, Dan He, Zi-Yang Yi, Di Zhao, Zhao Liu, Shui-Han Zhang, Lu-Qi Huang

**Affiliations:** 1School of Pharmacy, Changchun University of Chinese Medicine, Changchun 130021, China; rrzhou0823@163.com; 2National Resource Center for Chinese Materia Medica, China Academy of Chinese Medical Sciences, Beijing 100700, China; 3Hunan Academy of Chinese Medicine, Hunan University of Chinese Medicine, Changsha 410013, China; huangjianhua1985@163.com (J.-H.H.); hedan0915@126.com (D.H.); yiziyang1102@126.com (Z.-Y.Y.); didixiao0806@163.com (D.Z.); lz15243610938@163.com (Z.L.); 4Hunan Key Laboratory of TCM Prescription and Syndromes Translational Medicine, Hunan University of Chinese Medicine, Changsha 410208, China

**Keywords:** deep eutectic solvents, polysaccharides, ginsenosides, aqueous two-phase system

## Abstract

In this study, a green and effective extraction method was proposed to extract two main compounds, ginsenosides and polysaccharides, from American ginseng by combining deep eutectic solvents (DESs) with aqueous two-phase systems. The factors of type of DESs, water content in DESs, the solid–liquid ratio, extraction temperature, and extraction time were studied in the solid–liquid extraction. Then, the aqueous two-phase system (DESs-ethylene oxide–propylene oxide (EOPO)) and salty solution exchange (EOPO-salty solution) was applied for the purification of polysaccharides. The content of the polysaccharides and ginsenosides were analyzed by the anthrone–sulfuric acid method and HPLC method, which showed that the extraction efficiency of deep eutectic solvents (DESs) was better than conventional methods. Moreover, the antioxidant activities of ginseng polysaccharides and their cytotoxicity were further assayed. The advantages of the current study are that, throughout the whole extraction process, we avoided the usage of an organic reagent. Furthermore, the separated green solvent DESs and EOPO could be recovered and reused for a next cycle. Thus, this study proposed a new, green and recyclable extraction method for extracting ginsenosides and polysaccharides from American ginseng.

## 1. Introduction

Ginseng is one of the widely used medicinal materials with a history of use thousands of years old [1,2]. American ginseng (*Panax*
*quinquefolius* L.) is one specie in this system which has been harvested across eastern and central North America as well as being exported to China in abundance [3,4]. It has been utilized in functional foods, herbal medicine, and so on. American ginseng possesses various medicinal properties, such as antihyperglycemic [5,6], immunologic [7,8], cardiovascular [9,10], antioxidant [11,12], and anticancer effects [13], as well as improved neurocognitive function. Most of these medicinal effects are attributed to its main constituents, polysaccharides and ginsenosides [14]. The polysaccharides from American ginseng are natural polymers of more than 10 monosaccharides consisting of linear or branched carbohydrate chains joined by glycosidic linkages [15]. An extraction was performed for 25 similar polysaccharides from American ginseng using hot water or 0.3 M NaOH, and the structural characteristics of polysaccharides demonstrated that they have different molecular weights as well as different sugar components, consisting of Ara, Rha, Xyl, Man, Gal, Glc, GalA, and GlcA [16]. The CVT-E002 (Alexa Life Sciences, Inc., Edmonton, AB, Canada) is a patented natural product extracted from American ginseng. It contains 80% polysaccharides and 10% protein and is used for the prevention of acute respiratory infections [17]. As the main bioactive compounds in ginseng, the sum of ginsenosides Rb1, Rb2, Rc, Re, Rg1, and Rd were determined by HPLC-UV, which is not less than 4% for P. quinquefolius roots and 10% for the extracts according to U.S. Pharmacopoeia [18].

The conventional methods for extracting polysaccharides are based on hot water extraction and ethanol precipitation [19,20] while extracting ginseng saponins are based on certain concentrations of ethanol combined with heat-reflux, shaking, or ultrasound-assisted extraction [21,22]. These methods are unavoidable when using an organic reagent, complicated procedure, and long extraction time. Thus, establishing an efficient and green method for extracting polysaccharides and ginsenosides from American ginseng is important for its quality control, development, and applications.

Green solutions, such as ionic liquids (ILs) and deep eutectic solvents (DESs), have been developed and used in the extraction process to replace organic solvents [23,24,25]. As a new type of green extraction solvent, deep eutectic solvents (DESs) have the advantages of low cost, easy synthesis, good solubility, green degradability, etc., and have been widely used in many fields [26]. Recently, deep eutectic solvents have been reported to extract phenolic compounds in *Carthamus tinctorius* L. [27], flavonoids in Pollen Typhae [28], and polysaccharides in Camellia oleifera Abel. The aqueous two-phase system (ATPS) is a two-phase system in which two water-soluble substances are mixed under critical conditions; it is an efficient and gentle separation technique [29]. The combination with the aqueous two-phase system (ATPS) has increased possibilities for applications in natural plant extractions, for example, He et al. used aqueous two-phase systems (ionic liquids and salts) to extract ginseng saponins [30], the thermo separating polymers–DESs system and thermo separating polymers–salt-water system have been applied in the quick extraction and preliminary purification of polysaccharides [31,32,33]. The introduction of deep eutectic solvents into aqueous two-phase systems to study complex sample separation and analysis systems provides many new opportunities for the development of modern separation techniques.

There are many factors that affect the extraction rate of DESs, especially the effect of temperature, time, water content, and the solid to solvent ratio. Properly increasing the temperature can reduce the viscosity of DESs, increase the diffusion coefficient, and destroy the intermolecular interactions, thereby improving the dissolution of active substances. However, the higher temperature can also cause the cavitation effect of ultrasound, break some sugar chains and affect their biological activity. After a certain period of time, the dissolution rate of active ingredients increases with the increase in time, but after a certain period of time, the dissolution rate does not change much when the osmotic pressure of the solution system reaches equilibrium. The addition of suitable water can effectively reduce viscosity and can help perform better on polar compounds. However, excess water concentration leads to the loss of hydrogen bonds, which disrupts the structure of DESs. With the increase in the liquid–solid ratio, the medicinal powder is fully infiltrated, and the extraction efficiency increases continuously. However, when the solid–liquid ratio is too large, the sample and solvent cannot be fully mixed, because there are more solid particles in the solution, which may adsorb more solvent, resulting in a decrease in the extraction rate [34,35,36,37].

Therefore, taking the advantages of the two methods, this work aimed to establish a new and green method for extracting the two main compounds, ginsenosides and polysaccharides, from American ginseng by combining DESs with aqueous two-phase systems (ATPS). Crude American ginseng polysaccharides and ginsenosides extracts were obtained using DESs. The HPLC method was used to analyze the content of the ginsenosides for its quality assessment. Then the ATPS was designed and applied for the enrichment and preliminarily purification of the polysaccharides. The factors of type of DESs, water content in DESs, solid–liquid ratio, extraction temperature, and extraction time were studied in the solid–liquid extraction. The factors influencing the ATPS extraction, including the concentration of the ethylene oxide–propylene oxide (EOPO) copolymer and extraction temperature, were studied. Furthermore, the antioxidant activity and potential cytotoxicity of the polysaccharides was assessed.

## 2. Materials and Methods

### 2.1. Samples and Reagents

Panax quinquefolium was collected from Letaotao pharmaceutical company (Guangdong Province, China), and further identified and authenticated by the Hunan Academy of Chinese Medicine. All samples were dried to a constant weight, and grinded through the 100-mesh sieve. All the samples were kept at room temperature in the dark until analysis. Choline chloride, (>98.0%), ethylene glycol (>98.0%), butylene glycol (>98.0%), lactic acid (>95.0%), glycerol (>98.0%), formic acid (>88.0%), and anthrone (>98.0%) were purchased from Sinopharma Chemical Reagent Co., Ltd. (Shanghai, China). The standards Ginsenoside Rb1 (≥98%), Ginsenoside Rb2 (≥98%), Ginsenoside Rb3 (≥ 98%), Ginsenoside Rc (≥98%), Ginsenoside Rd (≥98%), and Ginsenoside Rg1 (≥98%) were purchased from Chengdu Chemical Reagent Co., Ltd. (Chengdu, China). H_2_SO_4_ (>98.08%) was purchased from Hunan Huihong Reagent Co., Ltd. (Changsha, China). HPLC-Grade Acetonitrile (Tedia, OH, USA) and phosphoric acid (Tianjin Hengxing Chemical Reagent Co., Ltd., Tianjin, China). were used for mobile phase preparation. Purified water was obtained from China Resources C’estbon Beverage Co. Ltd. (Changsha, China). Ethylene oxide–propylene oxide (EOPO) copolymer (Molecular weight of 2500) was purchased from Haian Co. Ltd. (Nantong, China). Ultrasonic cleaner (KM-500DB, 500 W) was purchased from Kunshan Meimei Ultrasonic Instrument Co., Ltd. (Kunshan, China).

### 2.2. Preparation of DESs

Four kinds of DESs were synthesized with the reported method [38]. Firstly, ChCl was selected as the HB acceptor and was mixed with ethylene glycol (ChEtgly), glycerol (ChGly), formic acid (ChFor), and lactic acid (ChLac) with a mole ratio of 1:2, respectively. Then the mixture solvents was stirred and heated at 80 °C for 30 min until homogeneous liquids formed. The detailed information of DESs is shown in Table 1.

### 2.3. Extraction Procedure

The extraction procedure is presented in Figure 1.

#### 2.3.1. Deep Eutectic Solvent Extraction Procedure

An accurately quantified 1 g of American ginseng powder and different kinds of DESs were added into an extraction vessel with ultrasonic extraction. For the screening of the extraction solvent, various types of DESs were compared based on the extraction yields of polysaccharides in American ginseng powder. The extraction parameters included water content in DESs (0%, 20%, 40%,60%, and 80%), extraction time (20, 30,40, 50, and 60 min), extraction temperature (40 °C, 50 °C, 60 °C, 70 °C, and 80 °C), and solid–liquid ratio (1:10, 1:15, 1:20, 1:25, and 1:30 mg·mL^−1^) were optimized using a single factor experimental design in order to maximize the extraction efficiency of the polysaccharides. Then, the crude DESs extracts were collected and centrifuged at 3000 rpm for 30 min. Following this, the supernatant was filtered through 0.45 μm nylon prior to the HPLC analysis of the ginsenosides and the precipitate was collected for subsequent polysaccharide determination.

#### 2.3.2. Aqueous Two-Phase Extraction Process (DESs-EOPO)

A certain amount of EOPO was added into the DESs tube. The mixture was well mixed until it formed an aqueous two-phase system with an EOPO-rich top phase and DES-rich bottom phase. The phase volume for the EOPO phase and DES phase was recorded and the polysaccharide concentration was determined, respectively.

#### 2.3.3. Aqueous Two-Phase Extraction Process (EOPO-Salty Solution)

The EOPO-rich phase was separated from the DESs-rich phase. Then, a monopotassium phosphate solution was added to the EOPO-rich phase and was left to stand for 16 h at 60 °C for the temperature-induced phase separation. The phase volume for the EOPO phase and salt phase was recorded and the polysaccharide concentration was determined, respectively.

#### 2.3.4. Dialysis

The polysaccharides in the aqueous phase were removed to a dialysis tubing (3500 Da) and dialyzed in distilled water for 24 h to remove salt and other small molecular impurities. The dialyzed solution was concentrated by rotary evaporation and dried in a vacuum oven at 50 °C to obtain the polysaccharides.

### 2.4. Conditions of HPLC Analysis

HPLC analysis was performed on a Shimadzu LC-20AT HPLC (Shimadzu, Japan) equipped with a binary solvent delivery bump, an auto sampler, and a Shimadzu SPD-M20A array detector (Shimadzu, Japan). Chromatographic separation was carried out on an Agilent Eclipse Plus C18 column (4.6 × 250 mm, 5 μm). The flow rate was 1.0 mL/min and the temperature was set at 30 °C. The mobile phase consists of acetonitrile (A) and 0.1% phosphoric acid aqueous solution (B). The gradient program was as follows: 0–34 min, 19.2% A; 34–35 min, 19.2–28.0% A; 35–48 min, 28% A; 48–56 min, 28–28.5% A; 56–71 min, 28.5–36% A, and they were filtered through a 0.45 μm membrane filter before use. The detection wavelength was set at 203 nm. The injection volume was 10 μL. All the samples were passed through a 0.45 μm membrane filter prior to injection.

### 2.5. Determination of Polysaccharides

The polysaccharide concentration in the crude DESs extract was determined by the common anthrone–sulfuric acid method. The determination procedure was as follows: 4 mL of pure water were added into the precipitate in Section 2.3.1 and were shocked until dissolved. The mixture was centrifuged at 2000 rpm for 30 min, then 0.1 mL of the supernatant was diluted to an ending solution volume of 10 mL with water, 1.0 mL of which was mixed with 4 mL of 0.2% anthrone–sulfuric acid solution. The mixture was allowed to stand in a boiling water bath for 20 min then immediately incubated at 0 °C for 10 min. The absorbance of the mixture was measured at 582 nm using the UV-visible spectrophotometry method. Results were expressed as mg/g of sample dry weight. Equation: Y = 0.0042X − 0.0003.

### 2.6. Antioxidant Activity of Polysaccharides In Vitro

The DPPH radical was widely used as a stable free radical to assess the free radical scavenging ability of antioxidants. The DPPH method involved the free radical, 2,2-diphenyl-1-picrylhydrazyl radical (DPPH). The polysaccharide solution was added to DPPH in an 80% methanol solution then diluted to 5 mL with 80% methanol. The mixture was shaken vigorously and kept for 30 min at ambient temperature (25 ± 0.2 °C) in the dark. The absorbance was measured in a 1 cm quartz cuvette at 515 nm against 80% methanol as a blank. A calibration curve was constructed using a series of standard ascorbic acid concentrations. All measurements were made in duplicate.

### 2.7. Cell Culture and Cytotoxicity Assay

The Raw 264.7 cell line was obtained from the cell resource center of the Shanghai Institute of Life Sciences, Chinese Academy of Sciences (Shanghai, China). The cell line was cultured in DMEM supplemented with fetal bovine serum FBS (10%), penicillin (10 U·mL^−1^), and streptomycin (100 g·mL^−1^) in a 5% CO_2_ incubator at 37 °C. Then, 100 μL of harvested Raw 264.7 was seeded in 96-well plates at 1 × 104 cell/mL and cultured overnight. Then, each cell line was treated using solutions with different polysaccharide concentrations and cultured for 24 h. The culture medium was added to 100 L MTT (0.5 mg·mL^−1^) per well for another 4 h. The culture medium was abandoned and 0.15 mL of DMSO was added. After shocking and mixing, the absorbance was measured under the wavelength of 490 nm on a microplate microscope to calculate the cell inhibition rate.

### 2.8. Statistical Analysis

All the experiments were carried out in triplicate. Data were expressed as the mean ± standard deviation (SD). Comparisons were performed by Student’s *t* test. *p* < 0.05 was considered statistically significant.

### 2.9. Method Validation

Linearity, precision, accuracy, and stability were used to evaluate the stability and feasibility of the method by the reported methods in the references [39,40]. Among them, the RSD value less than 3% was used as a standard to evaluate the stability of the method.

#### 2.9.1. Linearity

Working solutions containing seven ginsenoside standards were prepared with DESs and diluted to a series of appropriate concentrations to construct a calibration curve. The solution was passed through a 0.45 μm membrane filter and injected into HPLC for analysis.

#### 2.9.2. Precision

Take one sample of American ginseng, prepare the test solution according to the “Section 2.3.1” method, inject and analyze it according to the “Section 2.4” chromatographic conditions, and inject each sample six times continuously, and calculate the RSD of Rg1, Re, Rb1, Rc, Rb2, Rb3, and Rd.

#### 2.9.3. Accuracy

Take six American ginseng samples, prepare the test solution according to the “Section 2.3.1” method, inject and analyze according to the “Section 2.4” chromatographic conditions, and calculate the RSD of the peak areas of Rg1, Re, Rb1, Rc, Rb2, Rb3, and Rd.

#### 2.9.4. Sability (12 h)

Take one sample of American ginseng, prepare the test solution according to the “Section 2.3.1” method, inject and analyze it according to the “Section 2.4” chromatographic conditions, inject every two hours, and calculate the RSD of Rg1, Re, Rb1, Rc, Rb2, Rb3, and Rd. 3. Results and Discussion

## 3. Results and Discussion

### 3.1. Extraction and Separation of Polysaccharide form American Ginseng

#### 3.1.1. Optimizing the Extraction Conditions of DES

The whole extraction process contains two steps, first, the crude polysaccharides and ginsenosides were extracted by DESs, then the extractions were further divided into two parts. One part was used to determine the contents of ginsenosides directly, the other part was further purified by the aqueous two-phase system (DESs–EOPO) and salty solution exchange (EOPO–salty solution). Thus, in this section, we chose polysaccharide content extraction efficiency as the index for optimizing the DESs extracting conditions.

In this study, different types of DESs, the content of water in DESs, the solid–liquid ratio, extraction temperature, and extraction time were also optimized. The detailed results are listed in Figure 2.

(1)The components of DESs have a significant influence on its physic-chemical properties, such as polarity, viscosity, and solubility. They directly affect the extract efficiency of the target compounds. Four kinds of DESs were synthesized and optimized. ChEtgly obtained the best extraction effects for polysaccharide content (Figure 2A); therefore, ChEtgly was chosen as the extraction solvent for the following experiments;(2)The water content in DESs ranged from 10% to 50% were optimized. As can be seen from Figure 2B, the extraction efficiency increased when the water content in ChEtgly increased from 10% to 20%. When the water content was higher than 20%, the extraction efficiency decreased. Thus, 20% water content in ChEtgly was selected for this study;(3)The solid to solvent ratio will affect the diffusion of solute into the solvent, thus, in order to maximize extraction efficiency and reduce solvent waste, we studied the effect of solid to solvent ratios (1:10, 1:15, 1:20, 1:25, and 1:30 g/mL). As can been seen from Figure 2C, the best extraction effects for polysaccharide content was obtained with a 1:20 solid to solvent ratio;(4)The effect of extraction temperatures between 40 and 80 °C was investigated. It can be observed from Figure 2D that the extraction yield increases with the increase in temperature, then slightly decreases above a temperature of 60 °C, and the extraction yield was the highest when the temperature was 60 °C. Thus, the extraction temperature selected was 60 °C;(5)The effects of ultrasonic time ranging from 20 to 60 min on the extraction yield were also investigated, Figure 2E. A 30 min extraction time was the best extraction effect for polysaccharide content to be obtained.

#### 3.1.2. Optimizing the Extraction Conditions of Aqueous Two-Phase System

The aqueous two-phase system (DESs–EOPO) and salty solution exchange (EOPO–salty solution) method was applied for the purification of the polysaccharides. The purification process contained three steps, firstly, polysaccharides were first exchanged from DESs to EOPO, and then exchanged from EOPO to the salty solution, after that, polysaccharides were finally separated from the salty solution by a dialysis tubing.

For the first step, the ratio of EOPO to DES and the extraction temperature were also optimized. The detailed results are listed in Figure 3. The best conditions for the purification of the polysaccharides was a rate of 50% for EOPO to DESs and an extraction temperature of 40 °C. The maximum exchange efficiency (E.E.) of the polysaccharides was 91.2%. The polysaccharides were mainly extracted into the EOPO-rich phase during the first step.

Then, the EOPO-rich phase was further separated at the second ATPS. The EOPO phase rich with polysaccharides was mixed with monopotassium phosphate solution and left to stand for 16 h at 60 °C for the temperature-induced phase separation. The phase volume for the EOPO phase and salt phase was recorded and the polysaccharides concentration was determined, respectively. The extraction efficiency (E.E.) of the polysaccharides reached 87.02% in the second ATPS step. Thus, based on the two steps of purification, the polysaccharides were collected and obtained for further assessment.

### 3.2. Quantitative Analysis of Ginsenosides in DES Extracts

This section aimed to establish a fast method for determining ginsenoside contents for its quality control based on the DESs extraction and HPLC method. Under these optimized conditions, American ginseng samples were extracted by DESs. The contents of ginsenosides were further analyzed by HPLC. The HPLC chromatogram is shown in Figure 4.

#### 3.2.1. Validation of the Analysis Method

The working curves and other analytical performances, such as linearity, precision, and accuracy were used to evaluate the proposed method.

(1)Linearity: The working curves were drawn by plotting the peak areas value versus the concentrations of seven analytes in the spiked samples (Table 2);(2)Precision: The RSD of the peak areas of seven analytes were 0.78%, 0.56%, 0.65%, 1.01%, 1.22%, and 1.33%, respectively, all less than 3%;(3)Accuracy: The RSD of the peak areas of seven analytes were 0.82%, 0.61%, 0.65%, 1.35%, 1.30%, and 1.31%, respectively, all less than 3%;(4)Sability (12 h): The RSD of the peak areas of seven analytes were 1.35%, 1.21%, 0.79%, 0.88%, 1.25%, and 1.01%, respectively, all less than 3%.

#### 3.2.2. Comparison of Extraction Efficiency with Conventional Extraction Methods

The conventional method for extracting ginsenosides was based on the US Pharmacopoeia, which is extracted based on ethanol (40%) and 1-h heat reflux. As can be seen from Table 3, the contents for seven ginsenosides obtained by the DESs extraction method were higher than those obtained by the conventional method. The extraction effects for these ginsenosides were significantly higher than that by the conventional method. The total ginsenoside content for the conventional extraction method and DESs extraction method were 51.93 ± 1.02, and 71.59 ± 2.12, respectively. We compared the contents of seven ginsenosides extracted by the DESs extraction method with the existing literature data and found that the extraction efficiency of ginsenosides was significantly higher than that of conventional methods.

### 3.3. Antioxidant Activities of Ginseng Polysaccharides In Vitro

In the DPPH assay, the antioxidants were able to reduce the stable DPPH radical to the non-radical form DPPH-H. On the basis of this principle, the scavenging effects of ginseng polysaccharides at different concentrations on the DPPH radical are shown in Figure 5. The scavenging effects of ginseng polysaccharides on the DPPH radical increased concentration dependently and were 82.26%, 74.70%, 61.91%, 51.45%, 42.05%, 26.35%, and 16.76% at 20, 15, 10, 7.5, 5, 2.5, and 1.25 mg/mL, respectively, revealing that ginseng polysaccharides possess the highest DPPH radical scavenging activity (Figure 5).

### 3.4. Cytotoxicity of Assay In Vitro

The ginseng polysaccharides extracted by the conventional method and proposed method were selected to evaluate in vitro cytotoxicity using the RAW 264.7 cell line, as shown in Figure 6. The effect of ginseng polysaccharides on cell visibility showed a dose-dependent decrease with the concentration ranging from 0–100 ug/mL. Moreover, cell viability remained higher than 75. These results suggest the relatively low cytotoxic activity in vitro in the RAW 264.7 cell line (Figure 6).

## 4. Conclusions and Discussion

In this study, two main compounds, ginsenosides and polysaccharides, were extracted from American ginseng by DESs and aqueous two-phase systems, respectively. The crude American ginseng polysaccharides and ginsenosides were obtained by DESs in 30 min, which replaced the original 2-h heat reflux extraction method and obtained higher extraction efficiency. The separated green solvent DESs and EOPO were recovered and directly reused for a next cycle. The established DESs and ATPS extraction method has the merits of good environmental friendliness, high extraction ability, and simple operation, which can be a useful tool for the natural plant extraction.

The factors affecting the extraction rate of DESs, such as type of DESs, water content of DESs, solid–liquid ratio, extraction temperature, and extraction time, were investigated. It was finally determined that the extraction rate was the highest when the component of DESs was ChEtgly, the water content was 20%, the solid to solvent ratio was 1:20, the extraction temperature was 60 °C, and the ultrasonic time was 30 min. Moreover, the cytotoxicity test and the DPPH scavenging assay have been tested in the relevant literature on polysaccharide research [41,42]. Therefore, considering the further development and utilization of American ginseng polysaccharides, we have also tested it, indicating that American ginseng polysaccharides have no cytotoxicity and good antioxidant activity.

## Figures and Tables

**Figure 1 molecules-27-03132-f001:**
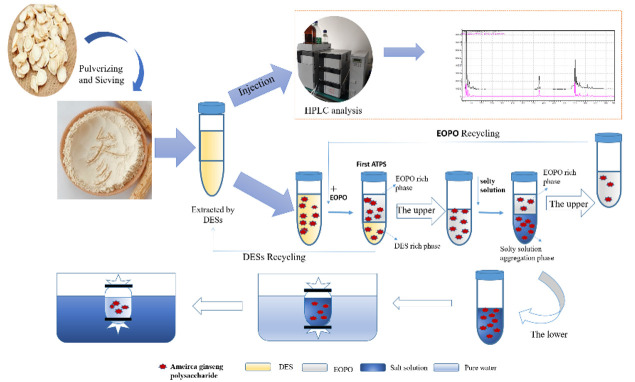
The representation of extraction procedure.

**Figure 2 molecules-27-03132-f002:**
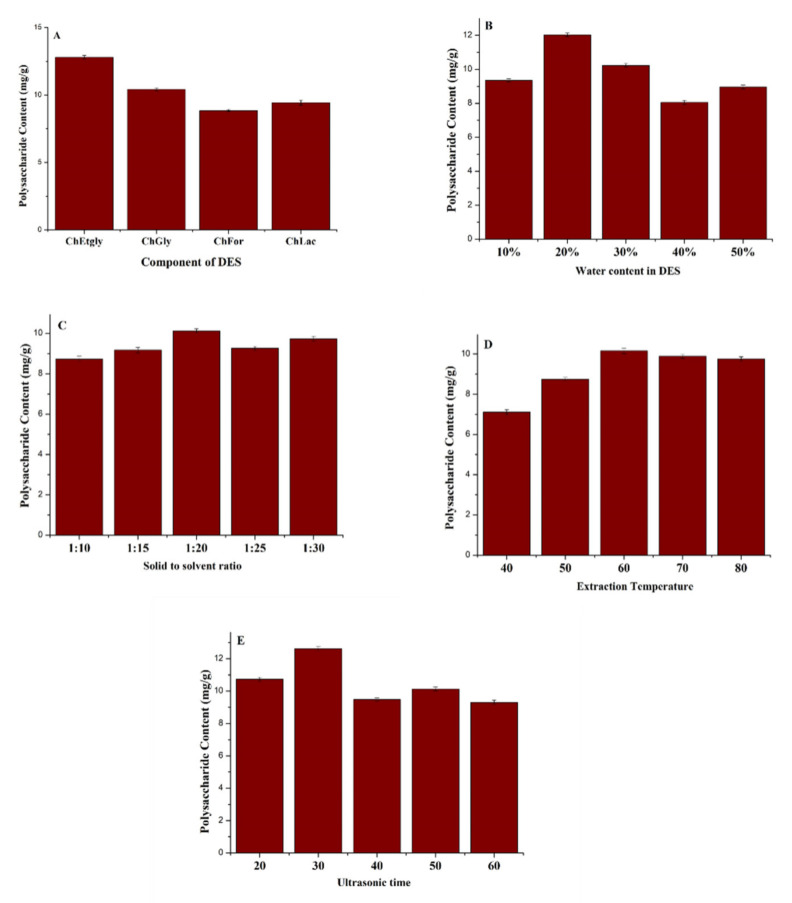
Optimized extraction process for DESs extraction (polysaccharide content (mg·g^−1^) at different DESs (**A**), water contents in DES (**B**), solid to solvent ratios (**C**), extraction temperatures (**D**), and extraction times (**E**)).

**Figure 3 molecules-27-03132-f003:**
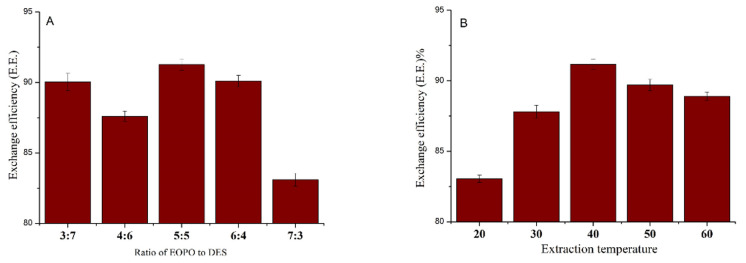
Optimized extraction process for ATPS extraction (exchange efficiency at different ratios of EOPO to DES (**A**), and extraction temperatures (**B**)).

**Figure 4 molecules-27-03132-f004:**
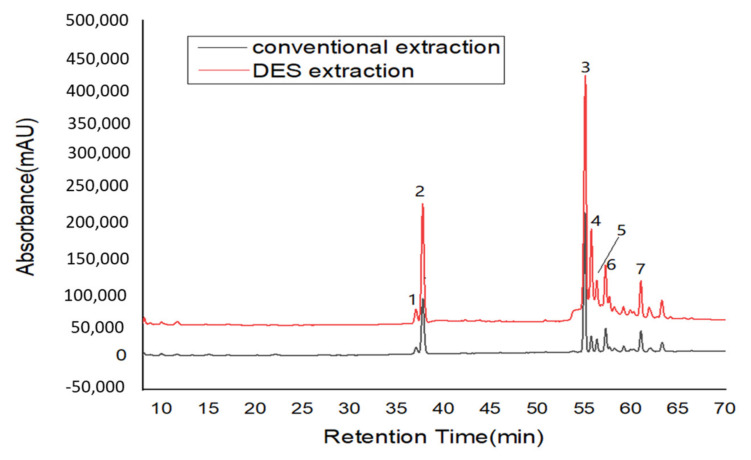
The HPLC chromatogram ginsenosides with different extraction methods (1: Rg1; 2: Re; 3: Rb1; 4: Rc; 5: Rb2; 6: Rb3; 7: Rd).

**Figure 5 molecules-27-03132-f005:**
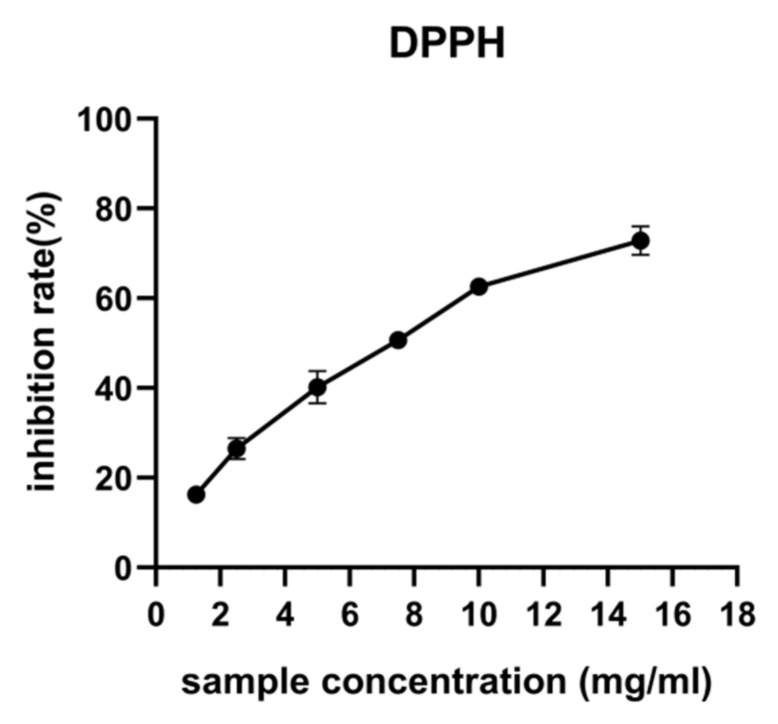
DPPH radical scavenging activity of ginseng polysaccharides.

**Figure 6 molecules-27-03132-f006:**
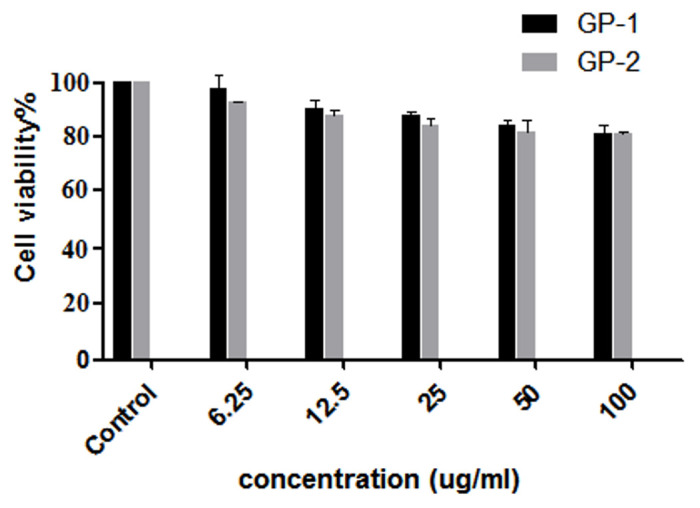
Effect of ginseng polysaccharides on cell viability.

**Table 1 molecules-27-03132-t001:** Information of synthesized DESs.

Hydrogen Bond Acceptor	Hydrogen Bond Donors	Name Abbreviation
Choline chloride	Ethylene glycol	ChEtgly
Choline chloride	Glycerol	ChGly
Choline chloride	Formic acid	ChFor
Choline chloride	Lactic acid	ChLac

**Table 2 molecules-27-03132-t002:** Regression equation for seven compounds.

Analytes	Regression Equation	Correlation Coefficient (r^2^)	Linear Range (mg·L^−1^)
Rg1	Y = 3369.5X + 3206.4	0.9997	50~200
Re	Y = 2774.8X + 31,440	0.9983	200~800
Rb1	Y = 2088.9X + 30,278	0.9994	500~2000
Rc	Y = 2350.8X − 8744.6	0.9978	125~500
Rb2	Y = 2452.8X + 4026.8	0.9981	125~500
Rb3	Y = 1972.8X + 6139.4	0.9997	125~500
Rd	Y = 1181.4X + 1207.6	0.9995	250~1000

**Table 3 molecules-27-03132-t003:** The contents of seven compounds by using different extraction methods.

Analytes	Conventional Method (mg/g)	DESs Extraction (mg/g)	*p*-Value
Rg1	1.34 ± 0.02	1.65 ± 0.02	4.54 × 10^−5^
Re	11.32 ± 0.52	14.15 ± 0.13	7.94 × 10^−4^
Rb1	26.89 ± 0.42	30.26 ± 0.33	3.98 × 10^−4^
Rc	1.84 ± 0.01	2.56 ± 0.02	6.19 × 10^−7^
Rb2	1.99 ± 0.02	3.58 ± 0.02	2.79 × 10^−4^
Rb3	4.57 ± 0.08	15.19 ± 0.09	1.10 × 10^−8^
Rd	3.98 ± 0.07	4.20 ± 0.05	1.14 × 10^−2^
Total contents	51.93 ± 1.02	71.59 ± 2.12	1.32 × 10^−4^

## Data Availability

The data presented in this study are available on request from the corresponding author.

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
