# Peer review of "Green and Efficient Extraction of Polysaccharide and Ginsenoside from American Ginseng (Panax quinquefolius L.) by Deep Eutectic Solvent Extraction and Aqueous Two-Phase System"

_molecules, 2022, doi:10.3390/molecules27103132_

Round 1

Reviewer 1 Report

The manuscript reports the efficient extraction of polysaccharide and ginsenoside from American ginseng using choline chloride-based deep eutectic solvents and an aqueous biphasic system. I found the results and discussion sections of the article to be lacking. Furthermore, the paper is not well-written, and the English is generally poor. As a result, I cannot suggest it for publishing.

  1. Page 1, line 37: Check the spelling of aitoxidants. Is it "antioxidants"?
  2. A more detailed explanation of the advantages of DES and two phasic systems in the extraction process should be included in the introduction.
  3. The extraction technique for ginsenosides and polysaccharides from American ginseng should be carefully explained (section 3.3).
  4. Section 3.2 states that four DES were prepared, yet only shows two DES (ChCl: EG and ChCl: BG). Figure 1a, on the other hand, makes no mention of the ChCl: BG. There is also a difference in the abbreviation of ChCl:EG in Figure 1a.
  5. The authors didn't pay much attention to the results and discussion, especially the effect of factors (Temperature, time, water content, solid to solvent ratio, etc.)?
  6. To yet, the authors haven't explained why they decided to use ChCl-based DESs for the extraction study?
  7. In abstract: reported that “Besides, the separated green solvent DESs and EOPO could be recovered and reused for a next cycle”. Why don’t the authors attempt to recycle the DES and EOPO after the extraction process?
  8. I would suggest the authors to compare and correlate the present study results with available literature data?
  9. The manuscript is riddled with grammatical and syntactical errors. Literally, the whole manuscript contains multiple errors with respect to word usage, grammar, or syntax. The entire manuscript must be carefully proofread before resubmission.

Author Response

The manuscript reports the efficient extraction of polysaccharide and ginsenoside from American ginseng using choline chloride-based deep eutectic solvents and an aqueous biphasic system. I found the results and discussion sections of the article to be lacking. Furthermore, the paper is not well-written, and the English is generally poor. As a result, I cannot suggest it for publishing.

  1. Page 1, line 37: Check the spelling of aitoxidants. Is it "antioxidants"?

Answer: Thanks for reviewer’s kindly advice, this mistake was corrected.

  1. A more detailed explanation of the advantages of DES and two phasic systems in the extraction process should be included in the introduction.

Answer: Thanks for reviewer’s kindly advice, we added more introduce about DES and two phasic systems in the extraction process to emphasize the advantages of current method. More detailed can be found in section 1.

  1. The extraction technique for ginsenosides and polysaccharides from American ginseng should be carefully explained (section 3.3).

Answer: Thanks for reviewer’s kindly advice, we added more explanation of the extraction process for ginsenosides and polysaccharides. After the American ginseng powder was added to DESs for extraction, the crude extracts of DESs were collected. After centrifugation, the supernatant was filtered through 0.45 μm nylon for HPLC analysis of ginsenoside content. And the remaining part was extracted with aqueous two-phase for polysaccharide determination. More detailed extraction process for ginsenosides and polysaccharides can be found in section 2.3.1, 2.3.2, and 2.3.3.

  1. Section 3.2 states that four DES were prepared, yet only shows two DES (ChCl: EG and ChCl: BG). Figure 1a, on the other hand, makes no mention of the ChCl: BG. There is also a difference in the abbreviation of ChCl:EG in Figure 1a.

Answer: Thanks for reviewer’s kindly advice, four types of DES used in this article was added in the revised manuscript. They are ChEtgly, ChGly, ChFor and ChLac. More detailed can be found in Table 1.

  1. The authors didn't pay much attention to the results and discussion, especially the effect of factors (Temperature, time, water content, solid to solvent ratio, etc.)?

Answer: Thanks for reviewer’s kindly advice, we added more discussion on the effects of temperature, time, water content, solid-solvent ratio and other factors on the extraction rate in the revised manuscript. More detailed can be found in section 4.

  1. To yet, the authors haven't explained why they decided to use ChCl-based DESs for the extraction study?

Answer: Thanks for reviewer’s kindly advice, previous literature has reported that ChCl-based DESs have better extraction efficiency. such as Huang et al.( DOI:10.13822/j.cnki.hxsj.2022008467) studied different types of DES and found that choline chloride is inexpensive, biocompatible, and low in toxicity, so ChCl -based DESs are the most widely used in extraction.

  1. In abstract: reported that “Besides, the separated green solvent DESs and EOPO could be recovered and reused for a next cycle”. Why don’t the authors attempt to recycle the DES and EOPO after the extraction process?

Answer: Thanks for reviewer’s kindly advice, the DES-rich bottom phase in 2.3.2 was collected for the next cycle and finally recycled.

  1. I would suggest the authors to compare and correlate the present study results with available literature data?

Answer: Thanks for reviewer’s kindly advice, we compared and correlated the present study results with available literature data. More detailed can be found in section 3.2.2.

  1. The manuscript is riddled with grammatical and syntactical errors. Literally, the whole manuscript contains multiple errors with respect to word usage, grammar, or syntax. The entire manuscript must be carefully proofread before resubmission.

Answer: Thanks for reviewer’s kindly advice, we carefully checked the word usage, grammar, and syntax of our article, and improved the quality of revised manuscript.

Reviewer 2 Report

The subject of the article is of interest, but the presentation for publication needs to be greatly improved.

The structure of the paper is inadequate and difficult to read and understand. A structure of type (1 Introduction, 2. Materials and methods, 3. Results and discussions, 4. Conclusions) is recommended.

The article is full of editing errors.

More information on the chemical structures of the polysaccharides and ginsenosides found in the American ginseng should be provided in the Introduction chapter.

No data (composition, eutectic temperature, viscosity, etc) are presented regarding the four kinds of DESs which were synthesized and optimized.

No data are presented regarding ultrasonic equipment (type, power, etc.).

In figure 1, figure 2 the units of measurement on axis are not specified.

Does Figure 1 refer to polysaccharide content or extraction efficiency? If it polysaccharide content then it must be expressed in mg / g. If it is efficiency then it is in percentage. How is the extraction efficiency calculated?

In Figure 5 the units on the axis for rate inhibition (%) are incorrect.

In figure 6 the axis for cell viability is in the range 0-100 (being percentages) not 0-120.

For the determination of polysaccharides by the anthrone-sulfuric acid method a calibration curve with a known carbohydrate (glucose, for example) was determined? If yes, then the results should be expressed in mg equivalent carbohydrate / g.

It is not clear whether only polysaccharides are found in the purified final extract or ginsenosides are also present.

Author Response

The subject of the article is of interest, but the presentation for publication needs to be greatly improved.

The structure of the paper is inadequate and difficult to read and understand. A structure of type (1 Introduction, 2. Materials and methods, 3. Results and discussions, 4. Conclusions) is recommended.

Answer: Thanks for reviewer’s kindly advice, we improved the structure(1 Introduction, 2. Materials and methods, 3. Results and discussions, 4. Conclusions) of article with more discussion and introduce the advantages of provide extraction method. The advantages of the extraction method are detailed in the section 1.  

The article is full of editing errors.

Answer: Thanks for reviewer’s kindly advice, we carefully checked the word usage, grammar, and syntax of our article, and improved the qualilty of revised manuscript.

More information on the chemical structures of the polysaccharides and ginsenosides found in the American ginseng should be provided in the 1 paragraph of Introduction chapter.

Answer: Thanks for reviewer’s kindly advice, we added more descriptions of polysaccharides and ginsenosides found in the American ginseng in the introduction chapter.

No data (composition, eutectic temperature, viscosity, etc) are presented regarding the four kinds of DESs which were synthesized and optimized.

Answer: Thanks for reviewer’s kindly advice, the information of synthesized DESs was added in Table 1. These four kinds of DESs were widely used, their composition, eutectic temperature, viscosity, etc information can be found in previous articles (DOI:CNKI:SUN:ZCYO.0.2020-17-025). 

No data are presented regarding ultrasonic equipment (type, power, etc.).

Answer: Thanks for reviewer’s kindly advice, the detailed information of ultrasonic equipment (KM-500DB, 500W) was added into the revised manuscript.  

In figure 1, figure 2 the units of measurement on axis are not specified.

Answer: Thanks for reviewer’s kindly advice, the units of measurement(mg/g)in figure 1, figure 2 was added.

Does Figure 1 refer to polysaccharide content or extraction efficiency? If it polysaccharide content then it must be expressed in mg / g. If it is efficiency then it is in percentage. How is the extraction efficiency calculated?

Answer: Thanks for reviewer’s kindly advice, Figure 1 refer to polysaccharide content, expressed in mg/g.

In Figure 5 the units on the axis for rate inhibition (%) are incorrect.

Answer: Thanks for reviewer’s kindly advice, we corrected the units on the axis for rate inhibition (%) in Figure 5.

In figure 6 the axis for cell viability is in the range 0-100 (being percentages) not 0-120.

Answer: Thanks for reviewer’s kindly advice, we corrected the axis range for cell viability in Figure 6.

For the determination of polysaccharides by the anthrone-sulfuric acid method a calibration curve with a known carbohydrate (glucose, for example) was determined? If yes, then the results should be expressed in mg equivalent carbohydrate / g.

Answer: Thanks for reviewer’s kindly advice, the results were also calculated, expressed in mg/g.

It is not clear whether only polysaccharides are found in the purified final extract or ginsenosides are also present.

Answer: Thanks for reviewer’s kindly advice, only polysaccharides were stayed in the purified final extract.

Reviewer 3 Report

The manuscript entitled "Green and efficient extraction of polysaccharide and ginsenoside from American ginseng (Panax quinquefolius L.) by deep eutectic solvents extraction and aqueous two phase system" presents an interesting issue but gives the impression that it was preliminary, incomplete research.
What could actually be done with the obtained results in terms of statistics (if the number of repetitions was appropriate) has not been done. However, it is unknown why scavenging of DPPH radicals and cell toxicity studies were performed. Moreover, these two experiments are not included in the Abstract nor Conclusions.
Detailed comments.
Page 3, line 98 - The Authors say that 1:25 solid to solvent ratio is the best, while the graph shows that 1:20.
Page 3, line 105 - A similar discrepancy is with the optimal extraction temperature. In the Discussion, the Authors mentioned 40°C, while the graph indicates rather 30°C, which is also the Authors' temperature in the Conclusions.
In Figures 1 and 2, there are no error bars and no statistically significant differences marked, and if there are none, there is no information about it anywhere in the text. Overall, the statistics of all results obtained leaves much to be desired. How can anybody do statistical calculations on triplicate?
Point 2.2.1 It is difficult to say that validation has been carried out. Table 1 shows only the results of linearity (and not complete). How were precision and accuracy made? Or the stability mentioned by the Authors. Stability of what?
In Table 2, there are no significant statistical differences marked. However, the Authors sum up that "The extraction effects for these ginsenosides were significant higher than that by conventional method". Using what test was this calculated??
In addition, the manuscript should be checked for typos and no spaces (multiple clusters of 2, 3 words).
However, first of all, it is not known what plant substance the Authors studied. Throughout the text, only the plant species is mentioned. You can assume the root, but that's just an assumption. Also, what does "all samples" mean in Material and methods? How many samples did the Authors have?

Author Response

The manuscript entitled "Green and efficient extraction of polysaccharide and ginsenoside from American ginseng (Panax quinquefolius L.) by deep eutectic solvents extraction and aqueous two phase system" presents an interesting issue but gives the impression that it was preliminary, incomplete research.
What could actually be done with the obtained results in terms of statistics (if the number of repetitions was appropriate) has not been done. However, it is unknown why scavenging of DPPH radicals and cell toxicity studies were performed. Moreover, these two experiments are not included in the Abstract nor Conclusions.

Answer: Thanks for reviewer’s kindly advice, we increased the description of the scavenging of DPPH radicals and cell toxicity of polysaccharide. More detailed can be found in section 2.6, and 2.7.

Detailed comments.
Page 3, line 98 - The Authors say that 1:25 solid to solvent ratio is the best, while the graph shows that 1:20.

Answer: Thanks for reviewer’s kindly advice, 1:20 solid to solvent ratio is the best, we corrected this mistake. More detailed can be found in section 3.1.1(3).

Page 3, line 105 - A similar discrepancy is with the optimal extraction temperature. In the Discussion, the Authors mentioned 40 â„ƒ, while the graph indicates rather 30 â„ƒ, which is also the Authors' temperature in the Conclusions.

Answer: Thanks for reviewer’s kindly advice, we corrected this mistake. More detailed can be found in section 3.1.1(5).

In Figures 1 and 2, there are no error bars and no statistically significant differences marked, and if there are none, there is no information about it anywhere in the text. Overall, the statistics of all results obtained leaves much to be desired. How can anybody do statistical calculations on triplicate?

Answer: Thanks for reviewer’s kindly advice, the results showed in Figure 1 and Figure 2 were the mean value of triplicate determine.

Point 2.2.1 It is difficult to say that validation has been carried out. Table 1 shows only the results of linearity (and not complete). How were precision and accuracy made? Or the stability mentioned by the Authors. Stability of what?

Answer: Thanks for reviewer’s kindly advice, we added information about precision, accuracy and stability. More details can be found in section 3.2.1(2) (3) (4).

In Table 2, there are no significant statistical differences marked. However, the Authors sum up that "The extraction effects for these ginsenosides were significant higher than that by conventional method". Using what test was this calculated??

Answer: Thanks for reviewer’s kindly advice, we corrected the description of the extraction results for conventional method and provided method. As can be seen from Table 3, the contents for seven ginsenosides obtained by DESs extraction method were higher than those obtained by conventional method, especially Rb2 and Rb3.

In addition, the manuscript should be checked for typos and no spaces (multiple clusters of 2, 3 words).

Answer: Thanks for reviewer’s kindly advice, we carefully checked these mistakes and made corrections.
However, first of all, it is not known what plant substance the Authors studied. Throughout the text, only the plant species is mentioned. You can assume the root, but that's just an assumption. Also, what does "all samples" mean in Material and methods? How many samples did the Authors have?

Answer: Thanks for reviewer’s kindly advice, the plant substance we use is the dried root of American ginseng, and we correct the description of “all samples” in section Material and methods.

Round 2

Reviewer 1 Report

The authors responded well to all the concerns I raised here. I recommend this article for publication in its current form. 

Author Response

The authors responded well to all the concerns I raised here. I recommend this article for publication in its current form. 

Answer: Thanks for reviewer’s kindly advice.

Reviewer 2 Report

The paper can be accepted for publication. Still minor corrections are needed.

row 70, To be corrected Carthamustinctorius L

row 115 bibliographic reference for reported method.

row 183 To be corrected against80%

row 203 To be corrected Extraction

row 209 measure unit for ultrasonic time

row 403 Conclusions are enough

Author Response

The paper can be accepted for publication. Still minor corrections are needed.

- row 70, To be corrected Carthamustinctorius L

Answer: Thanks for reviewer’s kindly advice, this mistake was corrected.

- row 115 bibliographic reference for reported method.

Answer: Thanks for reviewer’s kindly advice, we added a bibliography for reported method on row 115.

- row 183 To be corrected against 80%

Answer: Thanks for reviewer’s kindly advice, this mistake was corrected.

- row 203 To be corrected Extraction

Answer: Thanks for reviewer’s kindly advice, this mistake was corrected.

- row 209 measure unit for ultrasonic time

Answer: Thanks for reviewer’s kindly advice, this mistake was corrected.

- row 403 Conclusions are enough

Answer: Thanks for reviewer’s kindly advice.

Reviewer 3 Report

The Authors improved the manuscript, but still:
- there are no error bars in tables 1 and 2,
- the aim for the polysaccharide cytotoxicity test and the DPPH scavenging assay has not been added,
- there is no statistical analysis of the results.
Moreover, there is no discussion of the results, except for the single inserts. The text added to the Conclusions should instead appear in the Introduction (together with the literature sources where this information came from).
In addition, information on how to perform validation should be included in Materials and Methods. According to what guidelines were they carried out? If they were performed according to ICH HARMONISED TRIPARTITE GUIDELINE, VALIDATION OF ANALYTICAL PROCEDURES: TEXT AND METHODOLOGY Q2 (R1), among other things, “Linearity should be evaluated by visual inspection of a plot of signals as a function of analyte concentration or content. If there is a linear relationship, test results should be evaluated by appropriate statistical methods, for example, by calculation of a regression line by the method of least squares. (…) The correlation coefficient, y-intercept, slope of the regression line and residual sum of squares should be submitted.“

Author Response

The Authors improved the manuscript, but still:-  there are no error bars in tables 1 and 2,

Answer: Thanks for reviewer’s kindly advice, we added error bars in figure 2 and 3.

- the aim for the polysaccharide cytotoxicity test and the DPPH scavenging assay has not been added.

Answer: Thanks for reviewer’s kindly advice. The cytotoxicity test and the DPPH scavenging assay have been tested in the relevant literature on polysaccharide research. Therefore, considering the further development and utilization of American ginseng polysaccharide, we have also tested it, indicating that American ginseng polysaccharide has no cytotoxicity and good antioxidant activity. More detailed can be found in section 4.

- there is no statistical analysis of the results.

Answer: Thanks for reviewer’s kindly advice. We calculated p-values to test whether the conventional method and the DESs extraction are significantly different. More detailed can be found in Table 3.

Moreover, there is no discussion of the results, except for the single inserts. The text added to the Conclusions should instead appear in the Introduction (together with the literature sources where this information came from).

Answer: Thanks for reviewer’s kindly advice. we added discussion of the results, moved part of the text in the Conclusions to the Introduction, and added literature sources about this part of the text.

In addition, information on how to perform validation should be included in Materials and Methods. According to what guidelines were they carried out? If they were performed according to ICH HARMONISED TRIPARTITE GUIDELINE, VALIDATION OF ANALYTICAL PROCEDURES: TEXT AND METHODOLOGY Q2 (R1), among other things, “Linearity should be evaluated by visual inspection of a plot of signals as a function of analyte concentration or content. If there is a linear relationship, test results should be evaluated by appropriate statistical methods, for example, by calculation of a regression line by the method of least squares. (…) The correlation coefficient, y-intercept, slope of the regression line and residual sum of squares should be submitted.“

Answer: Thanks for reviewer’s kindly advice. we added information on how to perform validation in Materials and Methods. For the validation of HPLC, we referred to some relevant literatures. Linearity, precision, accuracy, and stability were used to evaluate the stability and feasibility of the method by the reported methods in the references. Among them, the RSD value less than 3% was used as a standard to evaluate the stability of the method. More detailed can be found in section 2.9.
